# Estimating the Postmortem Interval of Wild Boar Carcasses

**DOI:** 10.3390/vetsci7010006

**Published:** 2020-01-05

**Authors:** Carolina Probst, Jörn Gethmann, Jens Amendt, Lena Lutz, Jens Peter Teifke, Franz J. Conraths

**Affiliations:** 1Friedrich-Loeffler-Institut, Federal Research Institute for Animal Health, Institute of Epidemiology, 17493 Greifswald-Insel Riems, Germany; joern.gethmann@fli.de (J.G.); franz.conraths@fli.de (F.J.C.); 2Institute of Legal Medicine, Goethe-University, 60323 Frankfurt am Main, Germany; amendt@em.uni-frankfurt.de (J.A.); lutz@med.uni-frankfurt.de (L.L.); 3Friedrich-Loeffler-Institut, Federal Research Institute for Animal Health, Department of Experimental Animal Facilities and Biorisk Management,17493 Greifswald-Insel Riems, Germany; JensPeter.Teifke@fli.de

**Keywords:** wild boar carcass, decomposition, total body score, forensic entomology, postmortem interval, checklist

## Abstract

Knowledge on the postmortem interval (PMI) of wild boar (*Sus scrofa*) carcasses is crucial in the event of an outbreak of African swine fever in a wild boar population. Therefore, a thorough understanding of the decomposition process of this species in different microhabitats is necessary. We describe the decomposition process of carcasses exposed in cages. Trial 1 compared a wild boar and a domestic pig (*Sus scrofa domesticus*) under similar conditions; Trial 2 was performed with three wild boar piglets in the sunlight, shade, or in a wallow, and Trial 3 with two adult wild boar in the sun or shade. The wild boar decomposed more slowly than the domestic pig, which shows that standards derived from forensic studies on domestic pigs are not directly applicable to wild boar. The carcasses exposed to the sun decomposed faster than those in the shade did, and the decomposition of the carcass in the wallow took longest. To assess the state of decomposition, we adapted an existing total body scoring system originally developed for humans. Based on our studies, we propose a checklist tailored to wild boar carcasses found in the field that includes the most important information for a reliable PMI estimation.

## 1. Introduction

In the event of an outbreak of African swine fever (ASF), the time of disease introduction and the extent of the affected area need to be assessed. It is therefore necessary to estimate the time that has elapsed since the death of a wild boar (*Sus scrofa*) found in the field, i.e., the postmortem interval (PMI). For domestic pigs (*Sus scrofa domesticus*), comprehensive forensic research has been undertaken [1], but little is known about the decomposition process of wild boar, and no systematic guidelines for PMI estimation have so far been available.

In a previous study, 32 wild boar carcasses were exposed to vertebrate scavengers during different seasons [2]. The decomposition process proved to be highly variable, even in carcasses of similar size. In summer, large carcasses were fully skeletonized within a few days, whereas skeletonization took several months in winter [3]. Other studies showed that numerous factors, including seasonal and environmental changes, affect the time it takes until a carcass is skeletonized [4]. In an attempt to explain the variation in the rate of decomposition, the concept of accumulated degree-days (ADD) was introduced [5]. The advantage of using ADD is that this figure accounts for both the time and the amount of accumulated temperature which the carcass was exposed to.

There are numerous studies on the techniques that can be applied to estimate the time that has elapsed between the death of an animal and the discovery of its carcass. In the first weeks and during the insect active period (April–October in Germany), the most reliable method is forensic entomology, either by estimating the age of the oldest necrophagous insects on the carcass or by analyzing the succession patterns [6]. Entomological aspects have mainly been studied using the carcasses of domestic pigs, as they serve as models for humans [1]. For example, in central Europe, insect succession studies with domestic pigs have been performed in Germany [7,8], Austria [9] and Poland [10,11]. However, during winter and for longer PMIs, estimations based on entomological data are less precise than in summer. In addition, they require a comprehensive knowledge about the local species composition, seasonal patterns of succession, and development times.

Another useful parameter for PMI estimation is the soil underneath the carcass. It is known that a carcass has a significant effect on the biological and chemical characteristics of soil, e.g., on the pH and the structure of its microbial community [12]. Decomposing fluids and organic material from the activity of bacteria, fungi, protozoa, nematodes, insects, and scavengers enter the soil, where they alter the local concentrations of nutrients. This eventually results in the formation of a carcass decomposition island (CDI) [13]. CDIs can be still detected after several years and have been extensively studied in different mammalian species, including domestic pigs [14,15,16]. Some authors suggest that the chemical analysis of the CDI can help in estimating the PMI [17,18]. Other PMI estimation methods include microscopic, molecular, biochemical, microbial, and radionuclide-based methods [19], but they are of little practical use in the veterinary field. In any case, PMI estimation during an ASF epidemic in wild boar does not need to provide the level of accuracy required at a human crime scene.

Previous studies outlined five to six decomposition stages, namely fresh, bloated, active decay, advanced decay, dry, and remains [20]. However, later research showed that the borders between two successive decomposition stages cannot be defined unambiguously. Also, different body parts may represent signs of different decomposition stages at one point in time [11]. To take this mosaic nature of decomposition into account, current studies quantify the amount of accumulated decomposition that has taken place using the Partial Body Score (PBS) [21].

The aim of the study was to understand the course of decomposition of wild boar carcasses in different microhabitats. We therefore describe three decomposition trials. In Trial 1, we expected that the carcasses of a wild boar and a domestic pig would be similar in terms of decomposition and the sequence of necrophagous insects. In Trial 2, we presumed that a wild boar piglet exposed to sunlight decomposes faster than piglet carcasses in sheltered places (shade and wallow). In Trial 3, we hypothesized that the effect observed in Trial 2 (small carcasses) would be similar in large carcasses, and that an adult wild boar placed in the sun would decompose faster than one in the shade. Based on the results, we propose a standardized checklist to report wild boar carcasses applicable under field conditions that lists the most relevant data required for PMI estimation.

## 2. Materials and Methods

The trials were conducted on the Isle of Riems, in the southwestern part of the Bay of Greifswald at the Baltic Sea (45° 10′ 59″ N; 13° 21′ 50″ E). The island is about 1250 m long and 300 m wide, and lies 6 m above sea level. It is surrounded by wire fencing, so that there is no human disturbance, and only single foxes (*Vulpes vulpes*), badgers (*Meles meles*), wild boar, roe deer (*Capreolus capreolus*) and wild birds are sporadically present.

All seven animals (one domestic pig and six wild boar) were euthanized in late summer/autumn by means of a barbiturate overdose. The animal welfare officer of the Friedrich-Loeffler-Institut informed the authorities in charge of licensing animal experiments in the federal state of Mecklenburg–Western Pomerania about the study plan. The authority decided that the study did not represent an animal experiment. Nonetheless, the study design took ethical (humane killing of the pigs) and environmental issues into account. For comparability with forensic studies, all carcasses were placed in cages to prevent large scavengers from gaining access. The day of carcass exposure was counted as Day 1. Each carcass was regularly examined visually and physically, photographed with an Iphone 6 (Apple), and monitored by a digital camera (Seissiger Special-Cam 3 Classic, Würzburg, Germany). Each camera was installed on a bar 1.5–2 m above the carcass and set to take one photo every 5 min. Date and time were recorded automatically on each picture. Videos of the decomposition process of each carcass were produced using VirtualDub (GNU General Public License).

Except for one specimen (Trial 2, sun), all the carcasses were examined for at least one year. The weather data on an hourly basis were obtained from the meteorological station in Greifswald, supplied by the German Weather Service (http://www.dwd.de/cdc), including air temperature (°C) and relative humidity (%) 2 m above ground, precipitation (mm), wind speed 10 m above ground (m/s), and cloud coverage in eights (1/8). To determine the ADD, we first calculated the average temperature for each day. Since temperatures below 0 °C inhibit biological processes associated with decomposition [5,22], only the daily averages above 0 °C were then summed up for all days to determine the ADD at any given point in time.

### 2.1. Trial 1 (Domestic Pig vs. Wild Boar)

For Trial 1, two metal cages (size 2 × 1 × 1 m, mesh size 3 × 2 cm, covered with a grid, 5 × 2 cm mesh size) were placed 75 m apart from each other on a concrete surface. On 7 August 2017, soil from a mixed deciduous–coniferous forest located near Greifswald (pH 3.5–4.0) was excavated, transported to the island and filled into the cages up to a height of 1 m. On 11 August 2017, a domestic pig (female, 60 kg) (1domestic) and a wild boar (female, 80 kg) (1wild) were simultaneously euthanized and placed in separate cages. Both carcasses were examined every 1–2 days until Day 160, and every 7–8 days until Day 371, i.e., until only bare bones and desiccated skin remained from 1domestic. Carcass 1wild was examined until Day 804 (26 months). On 48 sampling days (until the skeletonization of 1domestic was complete), insect specimens of different developmental stages were collected directly on, in or underneath the carcass. For each carcass and each sampling day, the insect stages were collected into one 50 mL Falcon tube and further processed, following the guidelines of the European Association for Forensic Entomology [23]. In brief, immature stages of flies and beetles were immersed in hot (>80 °C), but not boiling, water for approximately 30 s. Then, the water was poured off through a sieve and the insects were stored in 96% denatured ethanol at 4 °C. The insects were morphologically identified to species, family or genus level using the current systematic literature and voucher specimens from the Institute of Legal Medicine Frankfurt am Main. The insect succession on the carcasses of 1domestic and 1wild was compared with regard to the duration of insect presence on the carcasses, the frequency of forensically relevant species and the overall species composition. To test the similarity of the sampled insect communities, the Sorensen coefficient (SS) was calculated using the equation(00000)
$$S_{S}=\frac{2a}{\left(2a+b+c\right)}$$
where *a* is the number of species common to both carcasses, *b* number of species unique to 1wild and *c* the number of species unique to 1domestic. The range of *S_s_* is from 0 to 1, where 1 indicates high and 0 low similarity. Species richness was defined as the number of species in a sample. To quantify the abundance of the different species, the samples were clustered in different groups from 0 to 4 (0 represents no specimen, 1 = 1–10 specimens, 2 = 11–100, 3 = 101–500 and 4 > 500).

### 2.2. Trial 2 (Piglets in Sun, Shade, Wallow)

For Trial 2, two metal cages (2 × 1 × 1 m, mesh size 4.5 × 4.5 cm, covered with a grid, mesh size 5 × 2 cm) and a plastic pool (122 × 122 × 30 cm; 337 L capacity), filled with earth and tap water (hardness 4, not chlorinated), covered with a grid, mesh size 5 × 2 cm, were used. On 6 September 2018, four piglets were simultaneously euthanized and immediately exposed at the respective study sites. Piglet 1 (male, 23 kg) (2sun) was placed in a cage in open grassland, exposed to solar radiation. Piglet 2 (male, 21 kg) (2shade) was placed in a cage in forested land between deciduous trees and shrubs in the shade. Piglet 3 (male, 23 kg) (2wallow) was placed in a pool, under deciduous trees and shrubs, in the shade, so that half of the carcass lay in a simulated swamp, and the other half above the surface. During the study, no water was added or exchanged, but leaves were removed twice. The three carcasses were spaced at a minimum distance of 50 m to avoid cross-interactions of necrophagous insects that might otherwise be attracted by neighboring study sites [24]. Carcass 2sun was monitored until skeletonization was complete, and carcasses 2shade and 2wallow for 1 year (until 7 September 2019). All three carcasses were examined every 1–3 days until Day 44 (18 visits). Carcasses 2shade and 2wallow were examined every five days average until Day 82 (8 visits) and at less frequent occasions until Day 408 (2shade; 15 visits) or Day 367 (2wallow; 24 visits).

### 2.3. Trial 3 (Adult Wild Boar in Sun vs. Shade)

For Trial 3, two metal cages (2 × 1 × 1 m, mesh size 4.5 × 4.5 cm) were used. In contrast to Trials 1 and 2, the cages in Trial 3 had no lid, so that, in particular, avian scavengers had access. On 19 October 2018, two female adult wild boar were simultaneously euthanized and immediately placed into the cages. One carcass (100 kg) (3sun) was exposed on grass, 20 m from the reeds at the shore, to direct solar radiation. The other one (83 kg) (3shade) was exposed on forested land between deciduous trees and shrubs, 10 m from the shore, in the shade. On 2 January 2019, the cage in the sun was unintentionally flooded with brackish water from the Baltic Sea for a 16-h period due to a high tide. Both carcasses were monitored for 1 year (until 18 October 2019). In Trial 3, we regularly weighed the carcasses. To this end, a bar with a hook was installed over the cages. For weighing, a scale was hung from the bar using the hook and the whole cage weighed. The empty weight of the cage was subtracted from the total weight to determine the weight of the carcass. Both carcasses were examined every 4 days on average until Day 39 (8 visits), and every 9 days until Day 335 (35 visits).

### 2.4. Scoring Decomposition

In all trials, the state of decomposition was quantitatively assessed using a Total Body Scoring (TBS) system tailored to the specific decomposition patterns observed in wild boar. Therefore, we adapted an existing TBS system originally developed for humans [25,26] and later modified for domestic pigs [27]. Four decomposition stages were distinguished (fresh, early decomposition, advanced decomposition, skeletonization). The PBS of the head (PBSH), trunk (PBST) and limbs (PBSL) were assessed independently, using the terminology of Moffatt et al. [21]. Each body part was assigned a score starting with one and increasing one point for each morphological change. The PBSH could be assigned 1–5 points (1 during early decay, and 2 each during advanced decay and skeletonization), the PBST 1–11 (1 in the fresh stage, 4 during early decay, and 3 each during advanced decay and skeletonization), and the PBSL 1–4 (1 each in the fresh stage and early decay and 2 during skeletonization). The highest scores of PBSH, PBST and PBSL were then added up to determine the TBS. For example, if the trunk displayed discoloration (2 points), bloating (3 points) and skin slippage (4 points) at the same time, the PBST was four. If it displayed skin slippage, but no bloating, it was also four. The lowest possible TBS was 1 + 1 = 2 (fresh) and the highest 5 + 11 + 4 = 20 (advanced skeletonization). Based on the observations made in a previous study [2], we modified and reduced the list of morphological changes to the most distinctive ones, because not all changes observed in humans or domestic pigs are applicable to wild boar. At the end of the study, the quality and practicability of the adapted TBS system was evaluated on a selection of photographs, showing each carcass in various stages of autolysis, heterolysis and decomposition, displaying either the entire body, parts of it, or the interior. All assigned PBS were collated into a spreadsheet to quantify the TBS for each recorded day and to relate them to the respective ADD.

## 3. Results

Since all trials were started in late summer/early autumn, flies were attracted within minutes after exposure. The first changes typical for early decomposition, advanced decomposition and skeletonization were observed 2–5 days, 3–44, or 11–433 days postmortem, respectively (Table 1). In all carcasses, the most prominent and rapid changes were observed during the early stage (bloating, purging of decomposition fluids, loss of bristles and decrease in biomass). Carcass 1wild decomposed much slower than 1domestic and did not reach skeletonization even after two years. Carcasses in the sun decomposed faster than the sheltered ones. Carcass 2sun skeletonized the most rapidly (TBS 19 on Day 44) and 1wild most slowly (TBS 16 on Day 804) (Figure 1). In the carcasses 1domestic, 2sun, 2wallow and 3sun, the head decomposed more rapidly than the trunk or limbs (Table 1). Appendix A displays the correlation between the assigned PBS and ADD. In all carcasses, the PBS followed a curvilinear pattern, indicating rapid decomposition during the early phase, followed by a slower decomposition rate as time passes. Appendix A summarizes the weather data. The carcasses were exposed to a total ADD and cumulative rainfall of 3708 °C and 600 mm (1domestic); 8608 °C and 1184 mm (1wild); 609 °C and 31 mm (2sun); 3902 °C and 442 mm (2shade and 2wallow); 3867 °C and 528 mm (3sun and 3shade), respectively.

### 3.1. Trial 1 (Domestic Pig vs. Wild Boar)

While carcass 1domestic underwent all stages of decomposition and reached a TBS of 20 (on Day 323) (Figure 2), the highest TBS 1wild reached was 16 (on Day 804), i.e., two years later, the trunk was still not skeletonized. Instead, a large proportion of a hard and crumbly substance (adipocere) had remained from the trunk (Figure 3). After one year, carcasses 1domestic and 1wild weighed 8 and 22 kg, respectively, i.e., they lost approx. 52 kg (87%) and 58 kg (73%) of their initial body mass.

During the fresh stage (PMI 1–2 days), no significant differences were found between the two carcasses. Both displayed rigor mortis, pale mouth mucosa, and clouding of the cornea more or less at the same time. On Day 2 (35 ADD), early decomposition started manifesting in both carcasses (the purging of decompositional fluids). The change in color on the abdomen was noticeable in 1domestic, while on 1wild, a color change was not clearly visible. On Days 5 and 6 (106 ADD), bloating of the abdomen started in 1domestic and 1wild, respectively (Appendix A). On Day 8 (144 ADD), masses of fly larvae were present in both carcasses at the natural body openings, and the first differences between 1domestic and 1wild were noticed. On Day 8, 1domestic reached the peak of bloating, while the first fly larvae started to penetrate the skin of the umbilical area and to open the abdominal cavity. On Day 9 (160 ADD), the outer layer of the skin of 1domestic separated from the dermis and the body started to deflate. After that, the decomposition rates between both carcasses began to diverge more obviously.

After 22 days (379 ADD), decomposition fluids of 1domestic were purging into the surroundings, i.e., the trunk displayed the first signs of advanced decomposition (TBS 11). After 29 days (481 ADD), a clear decrease in biomass was observed (TBS 13). The purging of body fluids led to the dehydration of the soft tissue, causing the skin to turn leathery. After 28 weeks (1527 ADD), the trunk of 1domestic reached the first stage of skeletonization (TBS 17) (Figure 2). On Day 225, 1domestic was unintentionally covered by a layer of sand due to a storm. After 11 months (3022 ADD), only bones and mummified skin remained (TBS 20).

In 1wild, debloating did not start until Day 33 (539 ADD). On Day 50 (760 ADD), soft abdominal tissues started to liquefy and 1wild reached the stage of advanced decomposition (TBS 10). On Day 123 (1338 ADD), a slight decrease in biomass was observed (TBS 11). Then, the skin turned hard and the abdominal tissues, including intestinal contents and abdominal fat, formed a hard and crumbly substance (adipocere). During winter 2017/2018, the whole carcass, including the skin and abdominal contents, were completely hard. During this time, no gross changes and no insect activity were observed. In summer 2018, the abdominal tissues turned soft again, and insects resumed their activity. The same occurred in winter 2018/2019 and summer 2019, respectively. After two years (7555 ADD), the head and limbs were skeletonized (dry skin and bristles adhered to the bones, PBSH 5 and PBSL 4), while the trunk had turned into a dry and hard substance (PBST 8) (Figure 3).

#### 3.1.1. Entomology: Species Richness and Diversity

Overall, we found 20 insect species of the order true flies (Diptera) and beetles (Coleoptera) (Table 2, Appendix A). Regarding flies, the family of blowflies was the most common with six species, five of which occurred on both 1wild and 1domestic. Regarding beetles, 10 species from six different families were present, of which three families were only found on 1wild. On 1wild, 18 species were found, whereas only 13 species were found on 1domestic. Despite the differences, the same forensically relevant species, frequently used to estimate a minimum PMI, were present on both carcasses. The only exception was the blowfly *Calliphora vicina*, which was only found on 1wild, but occurred in small numbers in total.

#### 3.1.2. Entomology: Succession

From Day 0 to Day 40, the larvae of blowfly species, i.e., *Lucilia ampullacea*, *Lucilia caesar*, *Lucilia sericata*, *Phormia regina*, were the dominant colonizers on both carcasses and occurred in large numbers (>500), especially from Day 10 to 25 (Figure 4). In a second colonization wave, the larvae of the small Piophilidae, *Piophila nigriceps*, were almost constantly on the carcasses until Day 173 and sporadically present until Day 364 (results not shown). This species occurred in larger numbers and more frequently on 1domestic than on 1wild. Therefore, differences between the carcasses were only visible in the number of specimens of this species, but not in the duration of their presence. Nevertheless, two blowfly species, i.e., *Calliphora vicina* and *Calliphora vomitoria*, showed differences in the succession pattern between 1wild and 1domestic. *C. vicina* exclusively colonized 1wild and was abundant from Day 10 to 18 and from Day 262 to 273. *C. vomitoria* was the most abundant species on 1wild and present almost throughout the decomposition process until Day 301, while it was less abundant and only sporadically present on 1domestic, where it was absent from Day 139 onwards. Overall, the duration of colonization differed between the two carcasses. The wild boar was colonized over a longer period.

### 3.2. Trial 2 (Piglets in Sun, Shade, Water)

The decomposition rates of the three carcasses differed considerably. Carcasses 2sun and 2shade followed a similar decomposition pattern, although Appendix A decomposed at a faster rate than Appendix A (TBS of 19 on Day 44 and 413, respectively). Appendix A decomposed more slowly than the carcasses in the terrestrial environment, and its most advanced TBS was limited to 17 (on Day 367, 3902 ADD), i.e., after one year, clean bones, but also a big piece of adipocere were left over. After one year, carcasses 2sun and 2shade weighed 1 kg and 2wallow weighed 3 kg, i.e., they had lost 22, 20, and 20 kg, respectively (96%, 95% and 87% of the initial body weight, respectively).

In carcass 2sun, maggot masses started to develop with the onset of the bloated stage (Day 4, 67 ADD). Most tissues decomposed within 6–12 days and, after 44 days (609 ADD), only bones, cartilage and small pieces of tissue remained (Figure 5).

In carcass 2shade, bloating started one day later (Day 5, 85 ADD), and the maggot quantity and activity was less pronounced as compared to 2sun. A clear CDI was only visible around the umbilical region, were inner organs and decomposition fluids purged out. After thirteen months (4469 ADD), only dry skin and bones remained (Figure 6).

During the first month, i.e., during the fresh, early and advanced stages of decomposition, half of carcass 2wallow was submerged, and half protruded above the surface. On the upper half, maggot activity showed very rapid progression during the first 44 days (609 ADD) (Appendix A). After the maggots migrated into the water and most of the soft tissues had liquefied, the carcass was completely submerged in water. Similar to 2sun and 2shade, after approx. one month the bones were skeletonized. Nevertheless, when we took the remains out of the water to assess them (at seven time points), we noticed that the abdominal fat and connective tissues had formed a hard and crumbling mass (adipocere) that did not change much and was still left over after one year (3872 ADD) (Figure 7).

### 3.3. Trial 3 (Adult Wild Boar in Sun vs. Shade)

Carcass Appendix A decomposed faster than Appendix A. In 3sun, the opening of the abdominal wall occurred on Day 19 (160 ADD; TBS 7) and eight months later, skeletonization was almost complete (TBS 18 on Day 244, 1798 ADD) (Figure 8). In 3shade, the abdominal wall remained closed for half a year. Necrophagous insects invaded the interior of the carcass through the mouth and anus, and therefore advanced decomposition was already ongoing when the abdominal wall was perforated (TBS 14). After one year (3817 ADD), 3shade displayed skeletonized bones (Figure 9), while the remaining abdominal tissues had formed a hard, dry substance (adipocere) (maximum TBS 18 was reached on Day 292 or 2694 ADD). In both carcasses, the skin had formed a desiccated shell around the whole trunk after six months, so that the carcasses seemed largely intact on their surface. However, if the abdominal flap was lifted, advanced decomposition was obvious (Figure 10). After one year, carcasses 3sun and 3shade weighed 24 and 24 kg, respectively, i.e., they had lost 76 kg (76% of initial body weight) and 58 kg (70%), respectively. Appendix A shows the weight loss curves.

## 4. Discussion

This study highlights the macroscopic differences in the decomposition process of a wild boar and a domestic pig, and of wild boar carcasses in different microhabitats. The study was designed to prove basic concepts of decomposition and related timelines. The results were obtained empirically and only on a small number of carcasses. Hence, decomposition studies with a higher number of individuals in different locations and under different environmental conditions with/without scavengers, as well as different (starting) seasons are necessary to confirm the results. However, our approach was sufficient to test some hypotheses and to formulate new ones. In particular, we could show that the decomposition process of wild boar differs from that of a domestic pig. Interestingly, wild boar carcasses may be highly resistant to degradation under specific conditions.

We believe that it is not yet possible to design an algorithm for a rapid, (semi-)automatic PMI estimation for wild boar carcasses in situ. Too many variables related to the carcass itself and its environment influence the decomposition rate and many of them are not yet fully understood. 

Among the intrinsic factors, it is known from the literature that body size and weight represent important factors [28,29]. Putrefaction is slower in newborns, and wounds accelerate the decomposition process [30], whereas previous freezing slows it down [31]. In this study, these factors were eliminated by euthanizing the animals by injection and exposing their carcasses immediately after death. However, possible effects of the barbiturate used for killing the animals must be taken into consideration. Drugs may have an influence on the odor cues of the carcass and on the development of fly larval and/or puparial stages [32].

Among the extrinsic factors, large scavengers [25,33] and weather, especially temperature, precipitation and humidity, are the most critical ones [26,34]. Similar to forensic studies conducted with domestic pigs, e.g., [7,8], we protected the carcasses from large scavengers in cages. Although we observed some insignificant activity of small birds and rodents (data not shown, but visible in the Appendix A), decomposition was mainly due to insect activity. The protection from scavengers allowed for comparability with forensic studies, but at the cost that one of the most important factors for decomposition was excluded. In Germany, different mammal and avian species have been observed scavenging on wild boar carcasses. Previous studies showed that decomposition can proceed at a completely different rate, even in the same geographical region, simply due to scavenging behavior [3]. As soon as scavengers have opened the abdominal cavity and started to remove soft tissues, the PMI can easily be overestimated. 

Another important variable is the type of microhabitat, e.g., sunlight or moisture [35] and soil type [36]. The study was conducted on an island in the ‘Greifswalder Bodden’ of the Baltic Sea in northeast Germany, i.e., the carcasses were exposed to the specific climatic and entomological conditions of this region. Although the temperature and wind trends were as usual, compared to other regions in Germany the temperature and the light intensity are usually lower, and the wind speed is higher. Unfavorable weather conditions may prevent or affect insect activity [37] and especially increased wind speed is known to reduce insect abundance [38,39,40]. Also, the nutrient-poor, sandy grassland and the presence of surface water, which are typical features of the island, may lower the local insect biomass significantly [41]. Nevertheless, the insect fauna of the carcasses can be regarded as typical in sequence of occurrence and composition. One reason might be the high adaptation to an ephemeral resource like carrion. Blowflies are particularly able to react very quickly to signals and stimuli such as decomposition odors, and, for important taxa like *Calliphora vicina,* it has even been shown that e.g., an increase in wind speed can initially lead to increased flight activity [42]. The differences in occurrence and abundance between the two blowfly species *C. vicina* and *C. vomitoria*, as well as the preference for wild boar by *C. vomitoria*, are striking and an exception in view of the otherwise rather consistent and typical species composition and succession. Regarding habitat preferences, *C. vicina* is mainly an urban blowfly, and competition or avoidance strategies might be an explanation, but this cannot be resolved in the present study.

Trial 1 showed that forensic knowledge obtained using domestic pigs cannot be transferred directly to wild boar. From early decomposition until skeletonization, the TBS remained lower in the wild boar. While only bones and desiccated skin were left over from the domestic pig, the wild boar had not fully decomposed until the end of the observation period. Although the anatomy between both is, in principle, comparable, there are important differences that may affect the decomposition process. Wild boar have thicker skin than domestic pigs, densely covered with bristles [43,44]. It is therefore not surprising that the skin remains intact for a longer time, thus retaining carcass moisture, protecting the interior from vertebrate scavengers, and possibly slowing down the activity rate of maggots. The skin also gives the body an intact appearance from the outside for a considerable time. A voluminous body with a largely intact skin suggests a low degree of decomposition, even when the inner tissues are already liquefied or decomposed. If the carcass is only assessed from the outside, the PMI may thus be over- or underestimated. Especially in adult animals and during winter, when the skin is particularly thick and tough, and the shoulder area is covered by a thick layer of rind and fat, the abdominal flap should be lifted to assess the carcass from the inside. 

The body composition of domestic pigs and wild boar differs in muscle pH, fat distribution, and essential amino acids [45,46]. Although both are omnivorous, their gut flora might be different due to different nutrition habits. There is circumstantial evidence that insects respond differently to the nutritive value of different animal species, by growing faster or slower or reaching different sizes in their carcasses [32]. It is not known if such differences exist between domestic pigs and wild boar, but if they exist, they may influence not just the decomposition but also the attraction of the respective carcasses.

Significant differences were not only observed between the wild boar and the domestic pig, but also between the carcasses in different microhabitats. Similarly to previous studies [12], we observed an early phase of slow, an intermediate phase of rapid and a late phase of slow mass loss. In carcasses 1wild, 2wallow and 3shade, a substantial amount of an extremely tenacious material resistant to degradation, called adipocere, was observed. Adipocere is a chemical form of mummification [47] and may slow the decomposition rate or arrest the decomposition process altogether [30]. Probably, the moist, oxygen-deficient conditions underneath the dehydrated, thick skin favored the formation of adipocere.

It has been suggested that animals infected with ASF often select deathbeds in cool and moist habitats [48]. Therefore, we included carcass 2wallow in the study design. Here, we found impressive maggot activity during the first few weeks. Later, however, the soft tissues formed a nugget of adipocere that did not decompose over the course of one year, maybe due to the lower temperatures in the wallow or the protection from insects the water offered. Previous studies have shown that water can increase or decrease the rate of decomposition depending on quantity, salinity, pH, and other factors [49,50]. This is of particular importance for carcasses found in water. They may be better conserved than those found on a dry surface [51], which might indicate the importance of ASF-contaminated carcasses in wet places as long-lasting potential sources of infection.

In ASF-affected regions, a common method of carcass disposal is burial, which does not protect the carcass from insects completely [52,53]. Therefore, in Trial 2 we also included a fourth piglet (female, 19 kg), buried it 80 cm below the ground surface and protected the tomb with a grid. It was planned to exhume it one year later. Interestingly, a scavenger was able to detect the dead piglet in its grave and removed it shortly after burial. No remains of the carcass could be found. In any case, previous research has shown that the rate of decomposition of buried carcasses is extremely variable and depends mainly on climate, temperature, the type of soil, and the depth of burial [54,55].

The detection of a wild boar carcass in the field is usually brought to the attention of the veterinary authority by a hunter or a citizen not familiar with ASF and existing precautions to prevent the spread of the disease. Such carcasses are often beyond the fresh stage (PMI more than 24 h), as it is the smell of putrefaction that usually enables localizing them. In an ASF-affected or high-risk region, the PMI should be estimated as quickly and accurately as possible. The PMI may give a clue on the latest date of ASF introduction into the region and may therefore be important for further decisions on disease control measures, e.g., the size of restriction zones. However, decomposition depends on many factors and evaluating the whole scene might be too complicated for a person without a forensic background. The PMI estimation should therefore be done in two steps. First, all relevant information that is locally available should be recorded (Figure 11). This information, together with retrospective temperature data, can then assist a qualified person to estimate a most probable PMI and a range for its minimum and maximum. As a general rule, the longer the PMI, the broader the range of the estimate [56]. To ensure that all relevant information is documented, we propose a checklist for wild boar carcasses found in the field that includes (I) the most important data required for a rough PMI estimation and (II) optional data that may increase the precision of the estimate (Figure 11).

The “optional information” section of the proposed form includes the adapted TBS scheme, i.e., a list of visual changes of the head, trunk and limbs, but without the scores to be assigned. The person performing the on-site assessment is supposed to mark in the empty boxes for the various options. With this information, a qualified person can later assign the respective points and estimate the TBS. In the modified TBS system, the number of achievable points is lower than in the original scheme (20 points instead of 35) (Appendix A). Unlike the domestic pig, the skin of a wild boar is covered with bristles. The different types of discoloration are therefore subsumed to a single feature “discoloration of the skin”, and the feature “skin displays a shiny/polished appearance (glossy marble)” is not applied. Instead, opening of the abdominal cavity is included as a characteristic of the late stage of early decomposition. Unlike humans, wild boar do not have a well-defined neck; the neck is therefore assessed together with the trunk. Purging of decompositional fluids is not differentiated between mouth/eyes/nose. Even in the advanced stages of decomposition, most bones may be covered by desiccated skin and therefore may not be visible. Therefore, the possible features of limb bones were reduced from ten to four.

The quality and practicability of the adapted TBS system was evaluated using photographs, since in real life, persons with pathological or forensic expertise will often not be able to visit the site before the carcass is removed. The use of photographs had limitations, e.g., some body parts or the interior of the carcass were not always visible. Moreover, information that could have been gathered through scent and palpation was not available. Obviously, the amended TBS scheme is a shortlist and may need to be further refined. However, it may serve as a guide for the most prominent features a decomposing wild boar carcass usually displays.

Insect activity has proven to be a reliable indicator for PMI estimation, so it is also taken into account. Previous studies showed that after hatching, blowfly larvae can skeletonize swine carcasses within a few days [58]. However, at least in temperate and cold climate regions, episodes of low temperatures are common in winter, so insect activity can be slowed down or even stop completely. Therefore, unlike existing TBS schemes, we assess insect activity in a way that is independent from the decomposition stage.

The checklist also includes the CDI, which is a result of decompositional fluids from the carcass and the activities and by-products of the necrophagous fauna. The size of the CDI is determined by the size of the carcass, maggot mass and soil texture [13]. Both the soil starting to get dark and sticky as well as the death of vegetation are assigned to the early stage of advanced decomposition, whereas plant growth in the CDI is assigned to the stage of advanced decomposition/skeletonization.

The activity of mammal and avian scavengers is one of the most important factors influencing the breakdown of a carcass. As soon as the abdominal cavity is opened, the rate of decay can be greatly increased, as scavengers rapidly remove large portions of tissues and disarticulate bones and scatter remains [59]. However, evidence for vertebrate scavenging may be difficult to assess, so it is included in the “detail section” of the checklist.

Combined with temperature data retrieved from the nearest meteorological station, the collected information should enable a qualified person to estimate the PMI and its measure of uncertainty, even if he or she was not at the site where the carcass was found. The required temperature data comprise the last days or months, depending on the decomposition stage of the carcass. If possible, temperature measurements on site can prove the validity of the data provided by the weather station.

Future research will have to show if an algorithm can be developed that allows a (semi-) quantitative estimate of the PMI with the data collected using the proposed checklist.

## 5. Conclusions

Standards derived from forensic studies on domestic pigs are not directly applicable to wild boar. The decomposition process of wild boar seems to be slower than in domestic pigs, probably due to their hard and thick skin (rind) covered with bristles. Especially in winter, it may take several months until a wild boar carcass is skeletonized. A significant amount of tissue may form adipocere, which is highly tenacious in the environment. Sunlight accelerates the decomposition process, while standing water may slow it down. In adult wild boar, the difference between sunlight and shade is not as obvious as in piglets, possibly because the skin protects the inner organs and soft tissues so effectively that environmental factors lose relative importance.

All PMI estimation methods are imperfect and get more imprecise as the PMI increases; thus, caution is needed when estimating the PMI of wild boar that died a long time ago. The proposed checklist might be a helpful tool for a first and quick PMI estimate.

## Figures and Tables

**Figure 1 vetsci-07-00006-f001:**
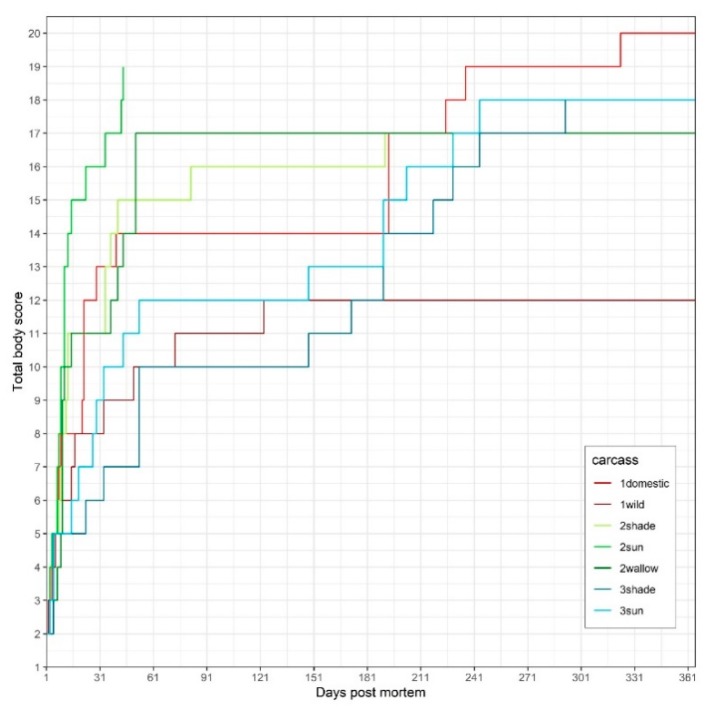
Total body score (TBS) of the seven carcasses of Trials 1 (1domestic, 1wild), 2 (2shade, 2sun, 2wallow) and 3 (3shade, 3sun) vs. postmortem interval (PMI).

**Figure 2 vetsci-07-00006-f002:**
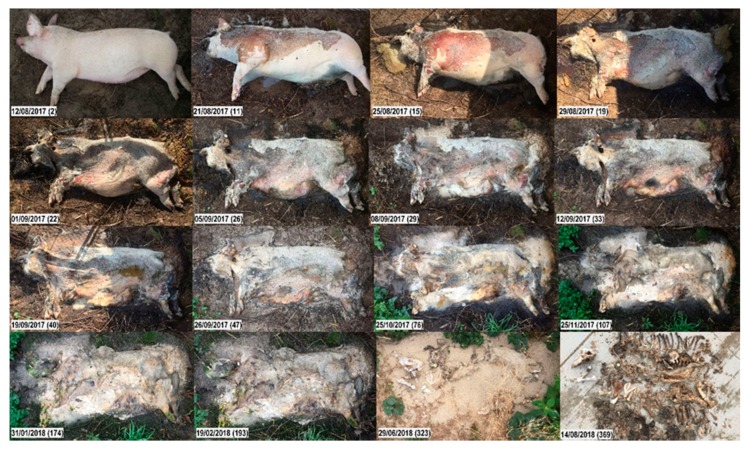
Decomposition of carcass 1domestic. Top to bottom, left to right; numbers in brackets represent the days post-mortem. Fresh. Day 2: Slight lividity, cooling of the body (algor mortis), clouding of the cornea. Early decomposition. Days 11, 15: The abdomen is open, gases have been released; carcass has reached the post-bloating stage; darkening of the skin, extensive skin slippage; purging of decompositional fluids from natural openings. Day 19: Head is in the advanced stage of decomposition. Advanced decomposition. Days 22, 26: Moist decomposition. Day 29: Decrease in biomass. The abdominal cavity collapses, leaving a flattened body; liquefied tissues continue purging into the surrounding. Day 33: Head already skeletonized (desiccated leathery skin, dry skull underneath). Day 40: Carcass has lost most of its soft tissues. Days 47–174: Moist decomposition. Skeletonization. Day 193: Most tissues other than bones have disappeared; bone exposure with greasy substance and decomposition tissues. Day 323: Carcass is covered by a layer of sand; underneath, dry bones and desiccated skin are left.

**Figure 3 vetsci-07-00006-f003:**
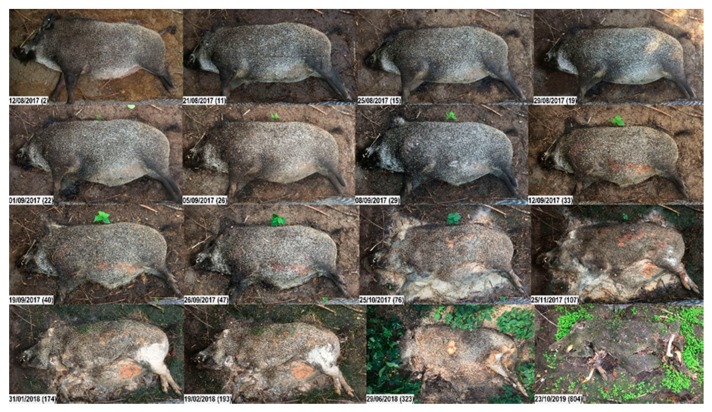
Decomposition of carcass 1wild. Top to bottom, left to right; numbers in brackets represent the days post-mortem. Fresh. Day 2: Cooling of the body (algor mortis), clouding of the cornea. Early decomposition. Days 11, 15: Abdominal cavity and proximal parts of limbs bloated; packages of fly eggs on the carcass; skin is intact. Days 19–29: Skin slippage while carcass still in the bloated stage. Day 33: Abdomen open, but skin still tight; maggots entered the abdominal cavity earlier, probably through natural openings (visible maggot movement inside the abdomen, see Appendix A). Days 40–47: Progressive skin slippage. Advanced decomposition. Days 76–107: Moist decomposition. Days 174–323: Large amounts of organic material including the skin have turned into a hard, soap-like substance (adipocere). Day 323: Carcass still hard; surface has dried. Skeletonization. Day 804: Exposed bones of limbs and head are dry and bleached. Skin is largely intact, although moss, fungus and small plants have grown on the surface during the last two years; note the vegetation growth on and surrounding the carcass.

**Figure 4 vetsci-07-00006-f004:**
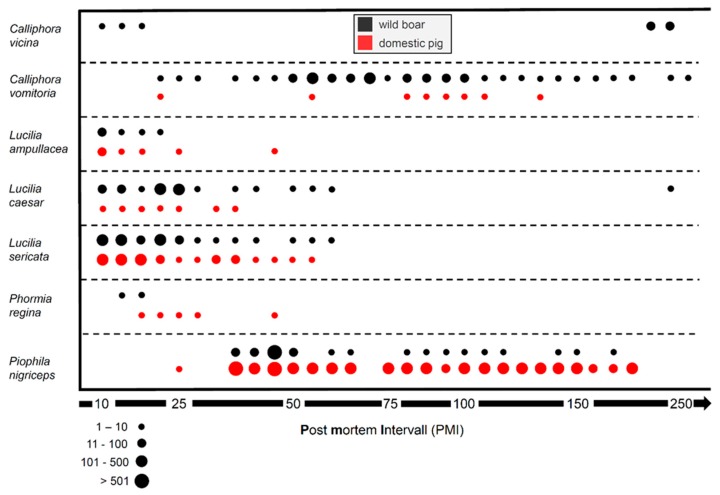
Insect succession on carcasses 1domestic (red) and 1wild (black) from Day 10 (20 August 2017) to Day 250 (17 April 2018) post-mortem. The abundance of the blowflies *C. vicina*, *C. vomitoria*, *L. ampullacea*, *L. caesar*, *L. sericata* and *P. regina* as well as of the piophilid *P. nigriceps* is shown by circles of different sizes representing the approximate number of specimens on the carcasses.

**Figure 5 vetsci-07-00006-f005:**
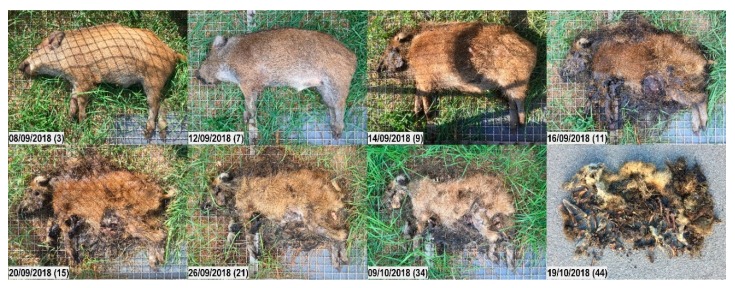
Decomposition of carcass 2sun. Top to bottom, left to right; numbers in brackets represent the days post-mortem. Early decomposition. Day 3: Still no visible swelling. Day 7: Extensive maggot activity on snout, eyes, and anus; moist decomposition of the head; abdomen bloated. Advanced decomposition. Day 9: Mummification of the head has started; extensive maggot activity over the whole body; abdominal tissues enter the stage of moist decomposition. Day 11: Skin has dried; head starts to skeletonize; reduced number of maggots; clear decrease in biomass. Day 15: Most soft tissues have gone; remaining tissues are black and sticky. Day 21: Further loss of soft tissues. Otherwise, not many changes are noticed. Skeletonization. Day 34: Bones easily palpable under the skin. Day 44: Dry bones, desiccated skin and bunches of sticky bristles left.

**Figure 6 vetsci-07-00006-f006:**
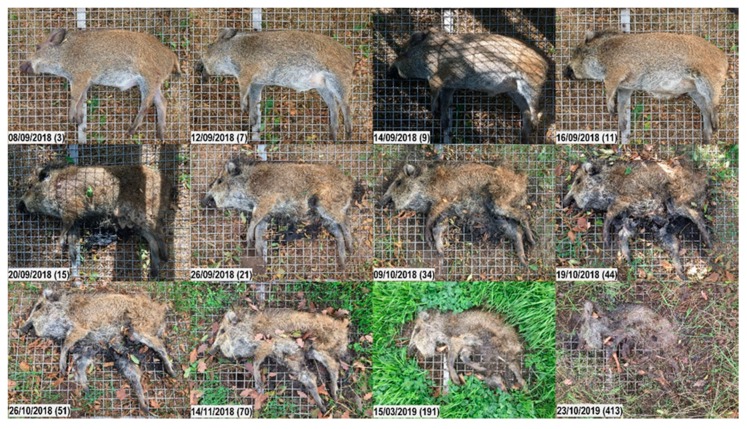
Decomposition of carcass 2shade. Top to bottom, left to right; numbers in brackets represent the days post-mortem. Early decomposition. Day 3: Still no visible swelling. Day 7: Moist decomposition of the head; abdomen bloated. Day 9: Head starts moist decomposition; Abdomen still not open. Day 11: Abdominal cavity is open. Advanced decomposition. Day 15: Head starts to mummify. Days 15, 21: Moist decomposition. Day 34: Decrease in biomass. Skeletonization. Days 44, 51, 70: Bones are prominent and easily palpable underneath the skin. Otherwise, not many changes are noticed. From Day 191: Dry bones, desiccated skin and bunches of bristles left. Note the vegetation growth surrounding the carcass.

**Figure 7 vetsci-07-00006-f007:**
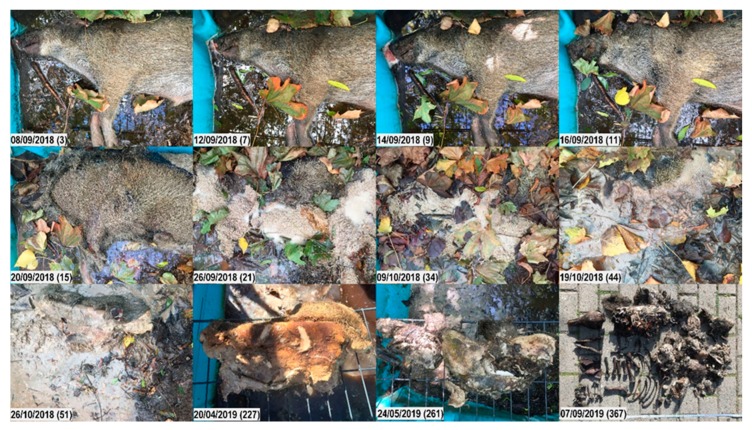
Decomposition of carcass 2wallow. Top to bottom, left to right; numbers in brackets represent the days post-mortem. Fresh. Day 3: Moist decomposition of the head has started. Day 7: Skin slippage on the head; decomposition fluids purge of mouth and nose. Early decomposition. Day 9: Blue discoloration of the abdominal skin; bloating or post-bloating not clearly visible. Advanced decomposition. Day 11: Massive maggot activity on the head; slippage of the abdominal skin. Day 15: Impressive surge in maggot activity on the whole carcass. Day 21: Maggot masses and foam cover the whole carcass; large numbers (thousands) of maggots migrate away from the carcass to pupate, and eventually drown in the water. Day 34: Reduced number of maggots; maggot activity has led to a large decrease in biomass; first exposure of bones due to colliquation of soft tissues. Skeletonization. Day 44: Skeletonized bones of head and limbs; soft tissues start to liquefy and dilute in the water. Day 51: Tissues continue liquefying. Day 227: While head and limbs are skeletonized, abdominal tissues have either liquefied or formed a soap-like adipocere. Day 261: The piece of adipocere has broken into several pieces, but otherwise has not changed much. Day 367: Carcass displaying a combination of skeletonization and advanced decomposition (adipocere).

**Figure 8 vetsci-07-00006-f008:**
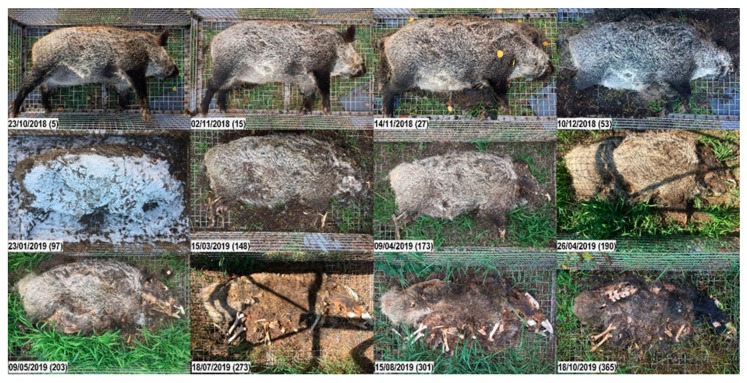
Decomposition of carcass 3sun. Top to bottom, left to right; numbers in brackets represent the days post-mortem. Early decomposition. Day 5: Bloated stage; small amount of decomposition fluids purging from mouth and nose. Day 15: Skin slippage and first loss of bristles. Day 27: Moist decomposition of the head; abdominal cavity open; massive loss of bristles over the whole carcass. Advanced decomposition. Day 53: First signs of bone exposure on the head; liquefaction of abdominal tissues; purging of decomposition fluids from the abdomen into the surrounding. On day 76, the carcass lay in brackish water for 16 h. Day 97: Carcass stiff, hard and covered with snow. Day 148: Bone exposure on the limbs (previously gnawed by rodents). Day 173: Skeletonization of the head progressing. Day 190: Notable decrease in biomass; bones of the head exposed, but still covered with cheesy substance. Note the vegetation growth surrounding the carcass. Day 203: Most soft tissues have disappeared. Skeletonization. Day 273: Bones of all body parts exposed, including the vertebra; some bones are clean and blank, others are still covered with cheesy tissue; most are disarticulated; skin is brittle and torn. Days 301, 365: Progressive skeletonization and dark discoloration of the skin; some bones still covered with greasy substance.

**Figure 9 vetsci-07-00006-f009:**
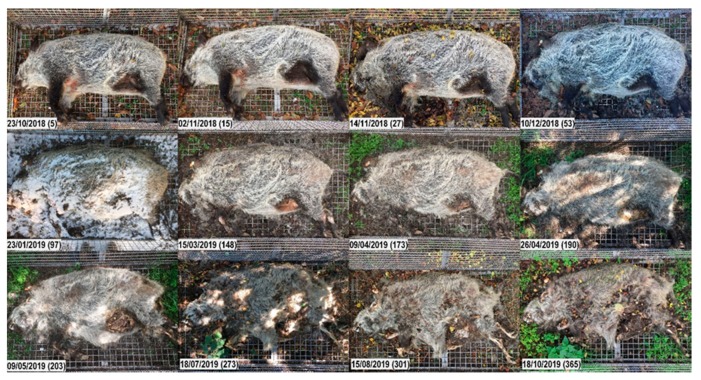
Decomposition of carcass 3shade. Top to bottom, left to right; numbers in brackets represent the days post-mortem. Early decomposition. Day 5: Bloated stage (less prominent than in carcass 3sun). Day 15: No big differences noted. Day 27: Skin slippage and massive loss of bristles; first opening on the neck/chest; purging of decomposition fluids. Advanced decomposition. Day 53: First signs of bone exposure on the head; even if the trunk looks intact and the abdominal wall is closed, the anal region is largely opened, and maggots have entered the abdominal cavity. Day 97: Carcass stiff, hard and covered with snow. Day 148: Bone exposure on the limbs (previously gnawed by rodents). Red fungus growing on the abdominal skin. Day 173: Mummification and skeletonization of the head progressing. Day 190: Decrease of biomass; the abdominal fat has been gnawed by rodents, but the abdominal wall is still closed; skin in the inguinal region is easily broken, when the leg is lifted. Day 203: Abdominal wall is opened from the inside (note that insects have already massively entered the abdominal cavity since the end of November). Day 273: Skeletonization of the head progressing. Skeletonization. Day 301: Although the carcass still looks well preserved, underneath the skin, bones of all body parts are skeletonized and covered with cheesy substance. Day 365: Carcass displays a combination of skeletonization and advanced decomposition. Remaining abdominal tissues have formed a hard, dry substance (adipocere).

**Figure 10 vetsci-07-00006-f010:**
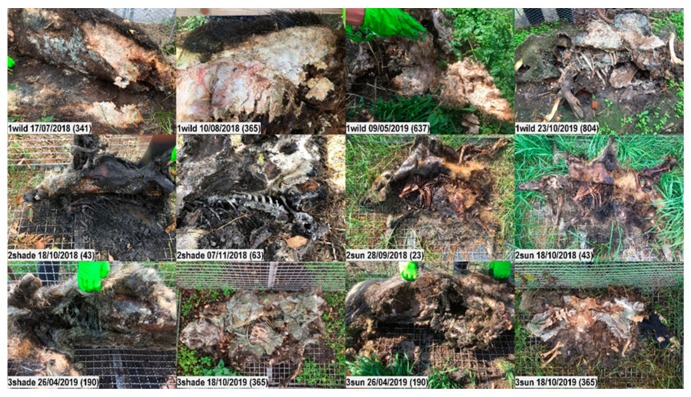
Insights into wild boar carcasses after lifting the abdominal flap. Top row: After more than one year, carcass 1wild still contains large amounts of hard and crumbly organic material underneath the almost intact and hard skin. On Day 341, the substance is dry, hard and crumbly; on Days 365 and 637, wet and bloody; on Day 804, the remaining substance is covered with fungus. Second row: Carcasses 2 shade and 2sun. Third row: Carcasses 3shade and 3sun.

**Figure 11 vetsci-07-00006-f011:**
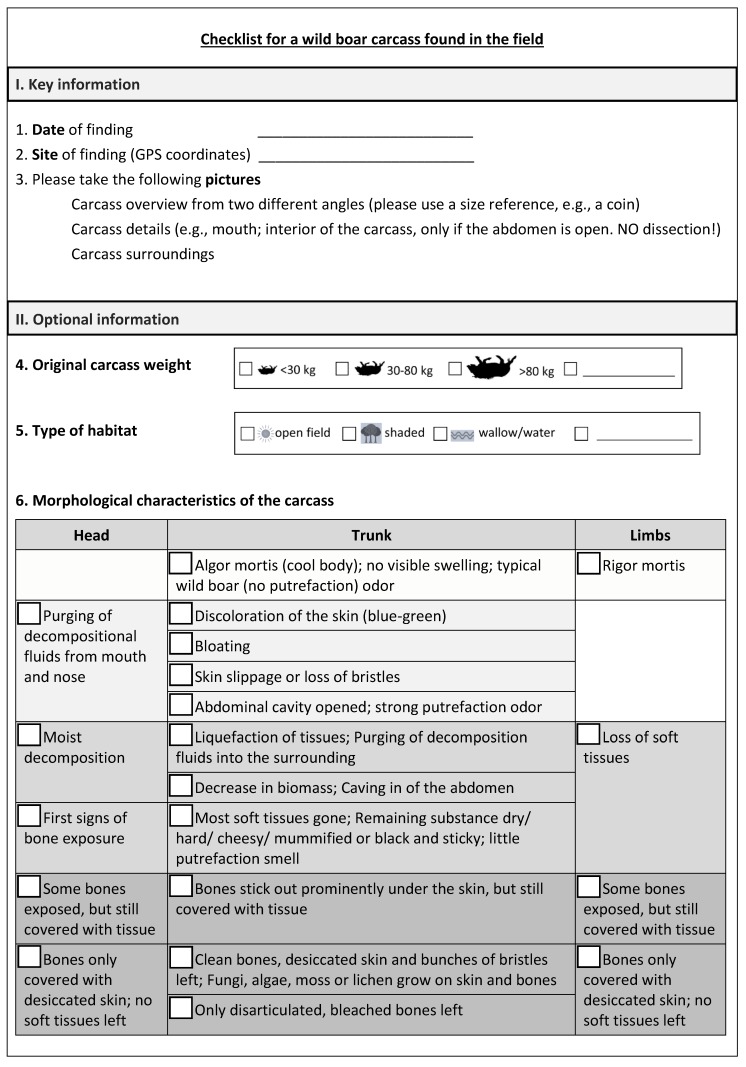
Proposed checklist for a wild boar carcass found in the field. The “key information” section includes the date and site of finding as well as the instruction to take photographs. Taking photographs is a simple and useful [57] technique for illustrating what the carcass looked like and where it was found, e.g., in the forest, near a road or a lake. The place of discovery cannot only give clues about the PMI, but also about the mode of disease introduction and spread.

**Table 1 vetsci-07-00006-t001:** Days post-mortem, on which the respective partial body score of the head (PBSH), trunk (PBST) and limbs (PBSL) points were assigned.

Carcass	Stage	Days Post-Mortem											Points Total
		Fresh	Early				Advanced			Skeletonization			
	Points	1	1	2	3	4	1	2	3	1	2	3	
1domestic	PBSH		2				8	22		21	193		5
	PBST	1	4	5	8	9	22	29	40	193	225	323	11
	PBSL	1					22			193	236		4
1wild	PBSH		2				8	15		433	708		5
	PBST	1	5	6	17	33	50	123	637				8
	PBSL	1					73			433	708		4
2sun	PBSH		3				7	9		11	34		5
	PBST	1	3	4	7	8	9	11	15	23	44		10
	PBSL	1					11			13	43		4
2shade	PBSH		3				9	13		34	413		5
	PBST	1	3	5	8	10	12	34	37	41	191		10
	PBSL	1					13			82	413		4
2wallow	PBSH		7				3	11		37	41		5
	PBST	1	9	n.o.	11	13	10	n.o.	51				8
	PBSL	1					15			44	51		4
3sun	PBSH		5				27	33		53	190		5
	PBST	1	5	3	15	19	44	190	203	244			9
	PBSL	1					29			148	229		4
3shade	PBSH		5				33	53		190	244		5
	PBST	1	5	5	23	190	53	190	218	292			9
	PBSL	1					173			148	229		4

**Table 2 vetsci-07-00006-t002:** Insect species found on carcasses 1domestic and 1wild.

Order	Family	Species	1wild	1domestic
Diptera	Calliphoridae	*Calliphora vicina*	x	
		*Calliphora vomitoria*	x	x
		*Lucilia ampullacea*	x	x
		*Lucilia caesar*	x	x
		*Lucilia sericata*	x	x
		*Phormia regina*	x	x
	Muscidae	*Hydrotaea ignava*	x	x
		*Muscina prolapse*	x	x
		*Ophyra capensis*		x
	Piophilidae	*Piophila nigriceps*	x	x
Coleoptera	Cleridae	*Necrobia ruficollis*	x	
		*Necrobia rufipes*	x	
	Dermestidae	*Dermestidae* sp.	x	
	Histeridae	*Histeridae* sp.	x	
	Nitidulidae	*Omosita colon*	x	
	Staphylinidae	*Aleochara* sp.		x
		*Creophilus maxillosus*	x	x
		*Omalium* sp.	x	
		*Philonthus* sp.	x	x
	Silphidae	*Necrodes littoralis*	x	x

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
