# Peer review of "Estimating the Postmortem Interval of Wild Boar Carcasses"

_vetsci, 2020, doi:10.3390/vetsci7010006_

Round 1
Reviewer 1 Report
Dear authors,
very well done,
a paper which is needed very urgently.
However, I have some minor remarks:
In general: This is just a small N, thus, your conclusions have to be drawn a bit more carefully. This is just a descriptive manauscript. As you describe yourselves, there is a very high variability. Thus also different locations (other study sites in Europe), higher numbers of individuals, diverse environmental conditions, with/without scavengers, and of course different (starting) seasons will have to be tested. Why did you expose the cadavers in late summer/autumn? Due to my experiences in the field I know that we have a very fastz start of decomposition in early summer (May/June). This would (perhaps) bring some other curves similar to 2sun. This is something which should be done in near future, and thus, should be also discussed.
Nevertheless, you have this one main finding: even after one year remains may contain ASFV. Thus there is always the possibility for a reinfection in a putative free area...
Some specific comments:
L87ff: Please add the start months also here.
L132: "four piglets"?? Here you only describe the exposition of 3 piglets. THe 4th buried one is not described in methods and in results, but you mention it in the discussion. Please doe describe that one also in methods and results!
L196: Tab 1: Please explain abbreviations in the caption (similar to Fig S1)
L199: Fig 1: explain abbr.
L263: please write "true flies" (lower case)
L264: name the scientific name of blowflies (Calliphoridae)
L271: Tab2: Although my origin is in systematics, I think we do not need all the names of the authors of first description here. Esp. as you did not name them in two of the species. Please delete the authors' names.
L287: very nice praph, would be even better, if you would show vertical dotted lines to show the months of the year (Aug, Sept...). Additionally here it gets clear, why we would need additional trials for different starting seasons
L404-410: well done!
L411ff: thus, we need more trials!
L431ff: True but in June we have within 1-3 days a "white body" due to eggs and larvea at least in yearling wild boar! (own field obeservations)
L464ff: Sus scrofa is one species! These are two forms of one species. Or perhaps you will have to cite, how you define a species (is there any differing vet-definition?) Please rewrite this paragraph!
L475: adipocere isntead of adiopocere!
L489"...we also included a fourth piglet..." Not described in methods and results! please do so!
L504: Ok, but this MS is just a first step into a very good direction! However, we need further trials starting in different seasons, differnt regions, different habitats, differnt conditions, with/without scavengers, ...
L549: This is a very important point! should also be discused a bit more detailed (and investigated in future)
kind regards
Reviewer 2 Report
The manucript from Probs et al., entitled "Estimating the postmortem interval of wild boar carcasses" described the decomposition process of swine and wild boar carcasses exposed in cages. Overall the results presented show low scientific soundness due to the fact that: - a very small number of animals was used in this study (1 domestic pig and 6 wild boars); - the animals were health, so the degradation of the carcasses can not be compared with free animals in natural conditions and/or with viral-infected animals; - the animals enrolled in the study were closed in a cage, protected from small carnovores not allowing to study the real scenario.
Reviewer 3 Report
In the current ASF epidemic in Europe and Asia transmission is mainly taking place in the wildboar-habitat epidemiological cycle. In general a lot is known about ASF epidemiology, but this is less true for concerning the wildboar-habitat epidemiological cycle. For this cycle substantial knowledge gaps still exist on aspects such as virus survival in the environment and in the carcasses. One important aspect there information is lacking is the decomposition process of wildboar carcasses. As more and more countries become infected, strive for control and eradication of the disease, and attempt to prove freedom, the aging of carcasses also become important in order to prove the date of the last infection. Therefore, this study brings very welcome, new, knowledge to the ASF research field.
The study is further well designed and very well written. I thoroughly enjoyed the reading, have only very few minor comments, strongly recommend the article for publishing in Veterinary Sciences, and congratulate the authors for a job well done.
Line 72-80: I suggest to start with the overall objective (aim) and after specify the specific aim of the 3 trials.
Line 131: Inpresume you mean “tap” water?
Line 147: I think that “lacked” is not the correct term here, they simply “did not have” a lid, I would suggest.
Line 154: I suggest to replace “boom” with “bar”.
Line 164: Please replace [21] with Mouffat et al (2106).
Line 223: I suggest to remove “(week 32)” as all other time points are given in days.
Line 227-238 and similar for the other figures: I suggest to change the layout for marking the decomposition phases (fresh and etc) in another way, it looks strange to have full sentences with just one word.
Line 244: Please define “hard as a stone” and “only little” in more exact (scientific) terms.
Line 489-494: This carcass and the reasons for its exclusion needs to be mentioned in material and methods.
Line 495.507: This seems to be introduction and not discussion (of the results in the study), please move it to the corresponding section.
Line 504-535: The development of the checklist and the procedure to follow on the detection of a carcass seems to be a result of the study? I suggest to move to result, and also include the development of the instruction and the form (i.e a standardized way to report carcasses) as an objective of the study.
Figure 11: Question 5 (habitat) sun/shade are not really habitat, but more related to weather, and the weather might of course have changed many times since the wild boar died. I presume what you are asking for is information whether the carcass lies in a place that allows exposure to sun. This needs to be clarified in the form. As one option is water, the others could be bush/shrubs or open field, or something along those lines.
Further down in the form you have one stage “bones easily palpable”. If the form is intended for the general public I suggest to include instructions NOT to touch the carcass.
Line 561: A space is missing behind the parenthesis.
Round 2
Reviewer 2 Report
The study still shows important scientific flaws, as pointed out in the previous revision report. The methology used doesnot support the conclusions presented.